# CORTICAL-SSM: A DEEP STATE SPACE MODEL FOR EEG AND ECoG MOTOR IMAGERY DECODING

## ABSTRACT

Classification of electroencephalogram (EEG) and electrocorticogram (ECoG) signals obtained during motor imagery (MI) has substantial application potential, including for communication assistance and rehabilitation support for patients with motor impairments. These signals remain inherently susceptible to physiological artifacts (e.g., eye blinking, swallowing), which pose persistent challenges. Although Transformer-based approaches for classifying EEG and ECoG signals have been widely adopted, they often struggle to capture fine-grained dependencies within them. To overcome these limitations, we propose Cortical-SSM, a novel architecture that extends deep state space models to capture integrated dependencies of EEG and ECoG signals across temporal, spatial, and frequency domains. We validated our method across three benchmarks: 1) two large-scale public MI EEG datasets containing more than 50 subjects, and 2) a clinical MI ECoG dataset recorded from a patient with amyotrophic lateral sclerosis. Our method outperformed baseline methods on the three benchmarks. Furthermore, visual explanations derived from our model indicate that it effectively captures neurophysiologically relevant regions of both EEG and ECoG signals. Our project page is available at `https://cortical-ssm-u90sg.kinsta.page/`

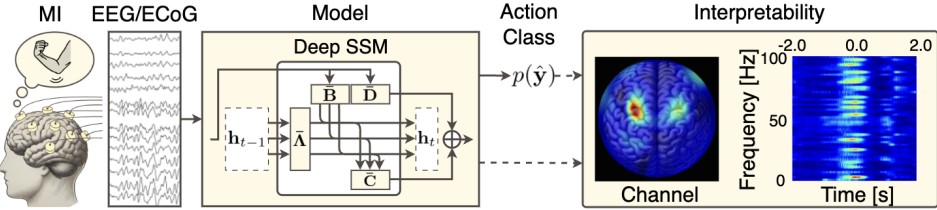

Figure 1: Task overview. The input is EEG or ECoG signals recorded while the subject imagines actions (e.g., elbow extension), and the model's output are predictions of the corresponding actions. The model also provides explicit interpretability across temporal, spatial, and frequency domains.

## 1 INTRODUCTION

Brain-Computer Interfaces (BCIs) hold transformative potential across various domains, including the diagnosis of neurodegenerative diseases, advanced brain function mapping, robotic control, and the development of immersive gaming devices (Wolpaw, 2013; Hramov et al., 2021). Among the various BCI paradigms, motor imagery (MI) BCIs decode intentionally modulated neural activity resulting from conscious cognitive effort, making them especially promising for developing assistive communication systems and neurorehabilitation protocols for patients with severe motor impairments (Hramov et al., 2021). Furthermore, BCI implementations leverage various techniques for recording brain activity, including electroencephalography (EEG), electrocorticography (ECoG), functional magnetic resonance imaging, and functional near-infrared spectroscopy (Ramadan & Vasilakos, 2017). Among these modalities, BCIs based on EEG and ECoG are particularly promising for real-world applications because of their high temporal resolution and superior portability.

In this study, we focus on brain activity occurring during motor imagery (MI) and tackle a classification task for imagined actions using both EEG and ECoG signals. Figure 1 presents a representative example of the task. In this case, the input comprises EEG and ECoG signals recorded while the

subject imagines an elbow extension. As its response, the model outputs predicted probabilities for each corresponding actions. Despite intensive research, the accurate decoding of MI EEG and ECoG signals is still non-trivial. For instance, binary classification task using MI EEG signals from the OpenBMI (Lee et al., 2019) dataset, a representative model (e.g., Lawhern et al. (2018)), demonstrates an error rate of approximately 22%.

Recent approaches have used Transformers to capture subject-agnostic dependencies in EEG and ECoG signals (Song et al., 2022; Wang et al., 2024a;d). However, to address the exponential growth in computational cost associated with longer time series, these methods patchify and compress the input EEG or ECoG signals along the temporal dimension, which may result in fine-grained temporal dependencies being lost. Moreover, while these approaches offer interpretability in spatio-temporal domains, they do not directly provide insights into the frequency domain, leaving the contributions of neurophysiologically significant frequency bands (e.g., mu band (Pfurtscheller et al., 2006)) unclear. Therefore, we propose Cortical-SSM, an extension of Deep SSM that captures integrated EEG and ECoG dependencies across temporal, spatial, and frequency domains. This design enables the modeling of temporal dependencies in EEG and ECoG signals without compressing them. Moreover, it provides direct visual explanations in each of the time, spatial, and frequency domains.

In Cortical-SSM, we extend a deep state space model (Deep SSM (Gu et al., 2022; Smith et al., 2023; Gu & Dao, 2024))—an architecture offering superior computational efficiency than Transformers for long sequence modeling—to effectively capture multi-scale temporal dependencies in EEG and ECoG signals. Specifically, we introduce the Frequency-SSM module, which extracts spatio-temporal dependencies for each frequency component, and the Channel-SSM module, which captures temporal-frequency dependencies for each electrode. Furthermore, we introduce a Wavelet-Convolution module integrating both deterministic and adaptive frequency features for feature extraction from the frequency-domain. The Frequency-SSM and Channel-SSM modules model temporal dependencies in EEG and ECoG signals without requiring the patchification employed in prior studies, thereby facilitating the capture of fine-grained temporal variations. Moreover, by incorporating the Wavelet-Convolution, Cortical-SSM extracts frequency-analyzable features while simultaneously learning their representations. Frequency-SSM and Channel-SSM modules explicitly model frequency-wise and electrode-wise features derived from the Wavelet-Convolution module, enabling direct visual explanations across temporal, spatial, and frequency domains.

Our main contributions lie in the following aspects:

- We propose Cortical-SSM, an extension of Deep SSM that captures integrated EEG and ECoG dependencies across temporal, spatial, and frequency domains.

- For frequency-domain feature extraction, we introduce Wavelet-Convolution, which integrates deterministically obtained frequency components with adaptively derived frequency features. This approach enables the extraction of interpretable features while preserving their learnable representations.

## 2 RELATED WORK

EEG and ECoG decoding methods have been extensively studied, as reviewed in (Abibullaev et al., 2023; Altaheri et al., 2023). Additionally, diverse predictive methods for multivariate time series signals have been systematically summarized in (Lara-Benítez et al., 2021; Liang et al., 2024). Furthermore, deep state space models (Deep SSMs) have emerged as a promising architecture for sequence modeling, with recent advances comprehensively documented in (Patro & Agneeswaran, 2024; Wang et al., 2024c).

**EEG and ECoG decoding.** Deep-learning based methods for decoding EEG and ECoG signals have been extensively investigated (Wang et al., 2024a; Song et al., 2022), and they demonstrate considerable potential for enhancing communication and rehabilitation in patients with physical paralysis (Abibullaev et al., 2023; Altaheri et al., 2023). Early attempts rely on convolutional architectures (Lawhern et al., 2018; Mane et al., 2021). Nonetheless, the limited receptive field of convolutional layers has motivated the exploration of Transformer–based approaches (Song et al., 2022; Miao et al., 2023; Jiang et al., 2024; Wang et al., 2024d) to model temporal dependencies in EEG and ECoG signals. Medformer (Wang et al., 2024d), for instance, segments input signals into patches of varying temporal lengths and then feeds these patches into a Transformer. These Transformer-based

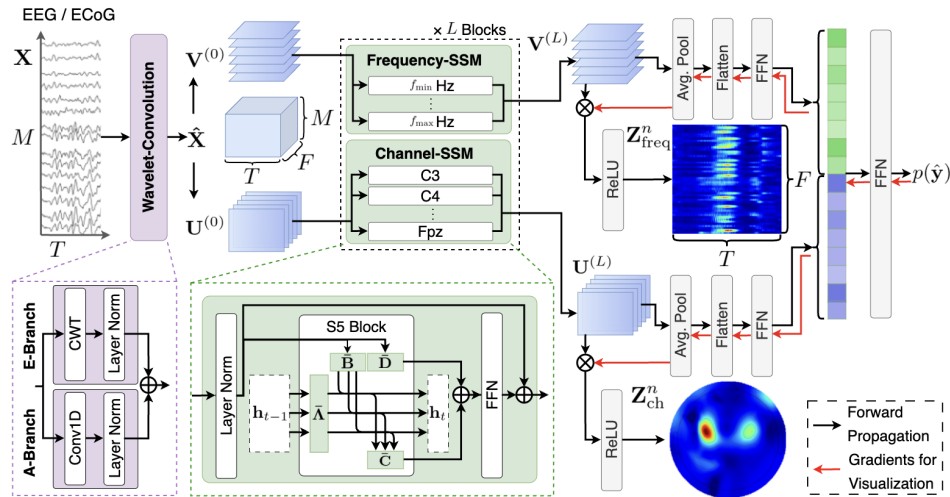

Figure 2: Overview of the proposed Cortical-SSM. Given an EEG or ECoG signal, the Wavelet-Convolution module extracts interpretable frequency features by combining deterministic and adaptive features (left). The Frequency-SSM and Channel-SSM then model spatio-temporal features for each frequency component and temporal-frequency features for each electrode, respectively (middle). Finally, the resulting features $\mathbf{V}^{(L)}$ and $\mathbf{U}^{(L)}$ are fused for action prediction, while providing visual explanations in spatio-temporal and temporal-frequency domains, respectively (right).

approaches typically employ temporal patching strategies prior to Transformer processing to address computational complexity in long-sequence modeling. However, this preprocessing step may disrupt fine-grained temporal dependencies in EEG and ECoG signals.

**Deep state space models.** Although the Transformer architecture has been widely adopted across various domains (Vaswani et al., 2017; Baevski et al., 2020; Dosovitskiy et al., 2021), the quadratic computational complexity $O(N^2)$ of its attention mechanism with respect to sequence length $N$ introduces fundamental scalability bottlenecks. This limitation has motivated extensive research into efficient alternative architectures (Ren et al., 2021; Han et al., 2023). Among these alternatives, Deep SSMs (Gu et al., 2022; Nguyen et al., 2022; Smith et al., 2023; Gu & Dao, 2024) have emerged as a promising framework for efficiently capturing long-range dependencies. In line with this trend, EEG classification models (Zhang et al., 2024; Tran et al., 2024; Gui et al., 2025) have also leveraged Deep SSMs to capture temporal dependencies in signals. However, current models predominantly adopt Mamba (Gu & Dao, 2024) as their foundational Deep SSM, leaving the optimal choice for the EEG or ECoG classification task ambiguous. In contrast, our proposed method extends S5 (Smith et al., 2023), a Deep SSM explicitly designed to capture inter-variable dependencies within the state space, to effectively address the intrinsic multivariate nature of EEG and ECoG signals.

Further related studies are discussed in Appendix A.

## 3 METHOD

### 3.1 CORTICAL-SSM

In this study, we address a classification task for EEG and ECoG signals recorded while subjects engage in motor imagery (MI) tasks. In this task, it is desirable to accurately classify the corresponding EEG or ECoG signals recorded while subjects engage in MI involving $N$ types of movement. Definitions of the key terminology used in this paper are provided in Appendix B. The input is EEG or ECoG signals recorded while the subjects performed MI, and the output is a predicted probability corresponding to the MI.

Unlike previous studies (Zhang et al., 2024; Tran et al., 2024; Gui et al., 2025), our method simultaneously captures the spatio-temporal features for individual frequency components as well as the temporal-frequency features of individual electrodes. This parallel modeling approach enables the integrated capture of spatial, temporal, and frequency dependencies, providing a more comprehensive representation for EEG and ECoG classification. Our approach, which is designed to capture

integrated dependencies among spatial, temporal, and frequency features, is widely applicable to the decoding of other endogenous and exogenous stimuli from EEG or ECoG signals.

Figure 2 shows the overall architecture of our proposed Cortical-SSM. The proposed method consists of three main modules, Wavelet-Convolution, Frequency-SSM, and Channel-SSM. The input $\mathbf{X} \in \mathbb{R}^{M \times T}$ to our model is the recorded EEG or ECoG signal, where $M$ and $T$ denote the number of electrodes and the sequence length, respectively.

### 3.2 WAVELET-CONVOLUTION

In the Wavelet-Convolution module, we integrate deterministically derived frequency components with learned frequency features. Previous models for EEG and ECoG classification widely employed one-dimensional convolutional layers to learn frequency features through their kernels (Lawhern et al., 2018; Song et al., 2022). Despite the promising results achieved by these models, their black-box nature poses challenges for direct interpretability and impedes frequency domain analysis of the extracted features. In contrast, the conventional short-time Fourier transform (STFT) and wavelet transform extract frequency components in a deterministic manner, thereby yielding inherently interpretable features. However, these fixed representations constrain the model's performance (see Subsection 5.3 for details).

To address this trade-off between learnability and interpretability, we propose a module integrating deterministically derived frequency components from an Explanation Branch (E-Branch) with trainable frequency features from an Adaptation Branch (A-Branch). This architecture maintains both the learnability of these representations and the extraction of non-black-box features. Specifically, we employ the continuous wavelet transform (CWT) for the E-Branch (first term of Equation 1), and a one-dimensional convolutional layer for the A-Branch (second term of Equation 1). Unlike existing methods that integrate the wavelet transform within convolutional layers (Fujieda et al., 2018; Williams & Li, 2018; Liu et al., 2019), our module processes CWT and 1D convolutional features in parallel, thereby enabling effective feature fusion.

Let $\mathbf{x}_m \in \mathbb{R}^T$ denote the EEG or ECoG signal recorded from the $m$-th electrode ($m = 1 \ldots M$). The module's output $\tilde{\mathbf{x}}_m \in \mathbb{R}^{F \times T}$ for each $\mathbf{x}_m$ is formulated as shown below, yielding $\tilde{\mathbf{X}} = \{\tilde{\mathbf{x}}_m \mid m = 1 \ldots M\} \in \mathbb{R}^{M \times F \times T}$:

$$\tilde{\mathbf{x}}_m = \frac{1}{2}\text{LayerNorm}\left(\text{CWT}\left(\mathbf{x}_m\right)\right) + \frac{1}{2}\text{LayerNorm}\left(\text{Conv1D}\left(\mathbf{x}_m\right)\right), \tag{1}$$

where $\text{Conv1D}(\cdot)$, $\text{CWT}(\cdot)$, and $\text{LayerNorm}(\cdot)$ represent one-dimensional convolutional layers, CWT, and layer normalization (Ba, 2016), respectively. Additionally, $F$ represents the dimension in the frequency domain. Details of the processing are described below.

First, in $\text{Conv1D}(\cdot)$, the kernel length $K$ is set to $K = f_{\text{sample}}/2$ following EEGNet (Lawhern et al., 2018), and frequency features are extracted. Here, $f_{\text{sample}}$ represents the sampling frequency.

In $\text{CWT}(\cdot)$, a filter bank $\mathbf{\Psi} \in \mathbb{R}^{F \times T}$ is constructed to extract features for individual frequency components. For the extraction of frequency components, the Morlet wavelet (Torrence & Compo, 1998) is employed as the standard mother wavelet. Let $\boldsymbol{\psi}_f = \left[\psi_{f,-\frac{T}{2}}, \psi_{f,-\frac{T}{2}+1}, \ldots, \psi_{f,\frac{T}{2}}\right] \in \mathbb{R}^T$ denote the Morlet wavelet used to extract a frequency component $f$. $\psi_{f,t}$ is defined as follows:

$$\psi_{f,t} = \sqrt{\frac{1}{s}}\pi^{-\frac{1}{4}}e^{i\omega_0 \frac{t}{s}}e^{-\frac{1}{2}\left(\frac{t}{s}\right)^2}, \ s = \frac{\omega_0 f_{\text{sample}}}{2\pi f}, \tag{2}$$

where $\omega_0$ and $s$ denote the center frequency and scaling factor, respectively. Based on the above, $\mathbf{\Psi}$ is expressed by the following equation:

$$\mathbf{\Psi} = \left\{\boldsymbol{\psi}_f \mid f = f_{\min} + \alpha(f_{\max} - f_{\min})/F, \ \alpha = 1 \ldots F\right\}, \tag{3}$$

where $f_{\min}$ and $f_{\max}$ denote the minimum and maximum $f$ targeted by the CWT, respectively.

Finally, we employ $\text{LayerNorm}(\cdot)$ to normalize the features obtained from $\text{Conv1D}(\cdot)$ and $\text{CWT}(\cdot)$. In this model, $\text{LayerNorm}(\cdot)$ is specifically applied along the temporal dimension, as explained below. Normalization techniques widely applied in deep learning include batch normalization (Ioffe & Szegedy, 2015), group normalization (Wu & He, 2018), and layer normalization across dimensions at a given time step. These methods normalize features across variables. However, in multivariate

time-series, when an arbitrary event occurs across variables and its effects appear at different time steps in the sequence, normalization across variables is known to introduce mutual noise (Liu et al., 2022; Kim et al., 2022; Liu et al., 2024). Similarly, in EEG and ECoG classification tasks, when internal stimuli derived from motor activity are recorded as signal sources via electrodes, these stimuli influence different temporal points within each electrode's signal. Therefore, we adopt layer normalization along the temporal dimension to address these issues.

### 3.3 FREQUENCY-SSM

The Frequency-SSM module independently captures spatio-temporal feature interactions within individual frequency component. Previous EEG and ECoG studies (Pfurtscheller & Neuper, 2001; Miller et al., 2007) have reported that MI tasks elicit frequency-specific power variations localized to functionally relevant cortical regions. Motivated by these neurophysiological findings, Frequency-SSM explicitly models the spatio-temporal dependencies for each frequency component in an independent manner. This design effectively tracks the power variations of those frequency bands associated with MI. The proposed module comprises $L$ hierarchically organized blocks, where each block includes layer normalization, a feed-forward network, and a Deep SSM. In the following, we provide a detailed description of the processing steps applied to the input $\mathbf{U}^{(l)}$ within the $l$-th block $(l = 0, \ldots, L)$, where $\mathbf{U}^{(0)} = \tilde{\mathbf{X}}$.

We first normalize $\mathbf{U}^{(l)}$ along the temporal dimension for each frequency component $f \in \{1, \ldots, F\}$:

$$\tilde{\mathbf{u}}_f^{(l)} = \text{LayerNorm}(\mathbf{u}_f^{(l)}). \tag{4}$$

Next, we capture the temporal dependencies in $\tilde{\mathbf{u}}_f^{(l)}$. In EEG classification tasks, Transformer-based approaches are widely employed to model temporal dependencies (Song et al., 2022; Wang et al., 2024d;a). However, for long-sequence modeling, Deep SSMs have demonstrated superior performance to Transformer (Gu et al., 2022; Gu & Dao, 2024). Therefore, our module incorporates a Deep SSM to capture temporal dependencies in EEG or ECoG signals. Deep SSMs can be broadly categorized into time-invariant (Goel et al., 2022; Smith et al., 2023) and time-varying (Gu & Dao, 2024; Dao & Gu, 2024) systems. Although time-varying Deep SSMs are prevalent in EEG classification (Zhang et al., 2024; Tran et al., 2024; Gui et al., 2025), prior work (Gu & Dao, 2024) suggests that their inherent selection mechanisms can be detrimental for continuous signals, and they have exhibited inferior performance in certain speech synthesis tasks. Therefore, we opt for time-invariant Deep SSMs. Furthermore, Deep SSMs can be classified by their input-output configuration into Single-Input Single-Output (SISO) (Gu et al., 2022; Dao & Gu, 2024) or Multi-Input Multi-Output (MIMO) (Smith et al., 2023; Soydan et al., 2024). Given the multi-electrode nature of EEG and ECoG signals, a MIMO configuration that preserves inter-variable dependencies within the state space is deemed appropriate. For these reasons, we extend S5 (Smith et al., 2023) as the Deep SSM with a time-invariant and MIMO configuration. The detailed mathematical formulation of S5 is provided in Appendix C.

By defining the operation of S5 as $\text{SSM}(\cdot)$, the feature $\mathbf{u}_f^{(l+1)}$ that captures temporal dependencies in $\tilde{\mathbf{u}}_f^{(l)}$ is obtained as the following equation:

$$\mathbf{u}_f^{(l+1)} = \text{FFN}\left(\text{SSM}\left(\tilde{\mathbf{u}}_f^{(l)}\right)\right) + \tilde{\mathbf{u}}_f^{(l)}, \tag{5}$$

where $\text{FFN}(\cdot)$ denotes the feed-forward network. Subsequently, the output $\mathbf{U}^{(l+1)} = \left[\mathbf{u}_1^{(l+1)}, \mathbf{u}_2^{(l+1)}, \ldots \mathbf{u}_F^{(l+1)}\right] \in \mathbb{R}^{M \times F \times T}$ of the module at the $l$-th block is obtained, capturing temporal dependencies independently for each frequency component.

### 3.4 CHANNEL-SSM

In the Channel-SSM, temporal-frequency features are extracted independently for each electrode. By explicitly modeling electrode-specific dependencies of temporal-frequency features, we capture localized variations in signal intensity associated with MI. This module comprises $L$ stacked blocks, each incorporating layer normalization, a feed-forward network, and Deep SSM layers hierarchically. The input $\mathbf{V}^{(l)}$ within the $L$-th block $(l = 0, \ldots, L)$ is modeled as follows, where $\mathbf{V}^{(0)} = \tilde{\mathbf{X}}$:

$$\mathbf{v}_m^{(l+1)} = \text{FFN}\left(\text{SSM}\left(\text{LayerNorm}\left(\mathbf{v}_m^{(l)}\right)\right)\right) + \text{LayerNorm}\left(\mathbf{v}_m^{(l)}\right), \tag{6}$$

yielding the block's final output $\mathbf{V}^{(l+1)} = \left[ \mathbf{v}_1^{(l+1)}, \ \mathbf{v}_2^{(l+1)}, \ \dots \ \mathbf{v}_M^{(l+1)} \right] \in \mathbb{R}^{M \times F \times T}$.

Finally, the outputs $\mathbf{U}^{(l)}$ and $\mathbf{V}^{(l)}$ from Frequency-SSM and Channel-SSM, respectively, are integrated as follows, yielding the predicted probability $p(\hat{\mathbf{y}})$ of action corresponding to $\mathbf{X}$:

$$p(\hat{\mathbf{y}}) = \mathrm{FFN} \left( \left[ \mathrm{AvgPool} \left( \mathbf{U}^{(l)} \right) ; \mathrm{AvgPool} \left( \mathbf{V}^{(l)} \right) \right] \right), \tag{7}$$

where $N$ represents the number of action types. Moreover, $\mathrm{AvgPool}(\cdot)$ denotes the average pooling layer, which aggregates the input features along the temporal dimension following the approach proposed in S4 (Gu et al., 2022). Furthermore, we use the cross-entropy loss as the loss function.

## 4 EXPERIMENTS

In the experiments, we used EEG and ECoG datasets recorded during MI tasks. For the EEG datasets, we used two public MI datasets OpenBMI (Lee et al., 2019) and Stieger2021 (Stieger et al., 2021). For the ECoG dataset, we used a clinical dataset collected from a subject diagnosed with amyotrophic lateral sclerosis (ALS; hereafter referred to as the ECoG-ALS dataset).

These datasets were selected for the following reasons. Previous research identified domain shifts across experimental sessions and subjects in EEG and ECoG signals recorded during MI tasks (Cho et al., 2017; Kaya et al., 2018). In the context of EEG, evaluating these two types of domain shifts is essential for developing reliable and broadly applicable models. Accordingly, we selected the publicly available OpenBMI (Lee et al., 2019) and Stieger2021 (Stieger et al., 2021) datasets, each of which comprise EEG signals recorded across multiple sessions from over 50 subjects. In contrast, ECoG signal acquisition poses substantial practical limitations because of the requirement for invasive craniotomy, making it infeasible to record signals from numerous subjects. Therefore, evaluating domain shifts across experimental sessions is the primary concern for ECoG. To this end, we selected a clinical dataset that consists of ECoG recordings made across eight experimental sessions from a single individual diagnosed with ALS. Furthermore, based on these properties, we adopted a cross-subject evaluation paradigm for EEG datasets (OpenBMI and Stieger2021) and a cross-session paradigm for the ECoG dataset (ECoG-ALS), employing $k$-fold cross-validation ($k = 8$).

For preprocessing, we followed the minimal approach reported by Delorme (2023). We downsampled EEG and ECoG signals to 250 Hz and used them directly as model inputs, without applying noise or artifact removal across electrodes. Further experimental details are provided in Appendix E.

## 5 RESULTS

### 5.1 QUANTITATIVE RESULTS

We conducted experiments to compare the performance of our model with baselines on the OpenBMI (Lee et al., 2019), Stieger2021 (Stieger et al., 2021), and ECoG-ALS datasets. The results for OpenBMI and ECoG-ALS are presented in Table 1, while the results for Stieger2021 can be found in Appendix F.1. Values reported in the table represent the mean and standard deviation obtained across $k$-fold cross-validation ($k = 8$). For the evaluation metrics, we employed accuracy, Macro-F1, AUROC (macro-averaged), AUPRC (macro-averaged), and Cohen's Kappa. We used these metrics because they are standard for the classification of EEG and ECoG signals.

This task involves (i) classification of EEG and ECoG signals, and (ii) handling of multivariate time-series signals recorded from multiple electrodes. Accordingly, we selected baseline methods from the following perspectives: (i) we chose baselines that have been successfully applied to EEG or ECoG classification, including ShallowConvNet (Schirrmeister et al., 2017), DeepConvNet (Schirrmeister et al., 2017), EEGNet (Lawhern et al., 2018), TSception (Ding et al., 2020), EEG Conformer (Song et al., 2022), and Medformer (Wang et al., 2024d), and (ii) given the multivariate nature, we selected baselines that have demonstrated effectiveness in multivariate time-series forecasting, such as Informer (Zhou et al., 2021), Autoformer (Wu et al., 2021), FEDformer (Zhou et al., 2022), Crossformer (Zhang & Yan, 2023), DLinear (Zeng et al., 2023), TimesNet (Wu et al., 2023), PatchTST (Nie et al., 2023), TimeMixer (Wang et al., 2024b), iTransformer (Liu et al., 2024), Uni-TS (Gao et al., 2024), TimeMachine (Ahamed & Cheng, 2024), and S-Mamba (Wang et al., 2025b).

Table 1 presents the performance of our proposed method, which achieved the highest accuracy, macro-F1, AUROC, AUPRC, and Cohen's Kappa on OpenBMI, with values of 81.62%, 81.56%,

Table 1: Performance comparison on OpenBMI (Lee et al., 2019) and ECoG-ALS datasets. **Bold** and underlined values indicate the best and second-best performances, respectively.

| Dataset | Models | Accuracy [%] ↑ | Macro-F1 [%] ↑ | AUROC [%] ↑ | AUPRC [%] ↑ | Kappa ↑ |
|---|---|---|---|---|---|---|
| | Chance Performance | 50.00 | 50.00 | 50.00 | 50.00 | 0.00 |
| | General Time-Series Models | | | | | |
| | Informer | 73.92±5.19 | 73.71±5.25 | 83.06±5.79 | 82.77±5.75 | 0.48±0.10 |
| | Autoformer | 67.91±4.84 | 67.73±4.94 | 74.80±5.79 | 74.19±6.04 | 0.36±0.10 |
| | FEDformer | 66.67±4.83 | 66.53±4.85 | 72.80±6.71 | 72.02±7.22 | 0.33±0.10 |
| | Crossformer | 71.85±4.93 | 71.74±4.99 | 80.05±5.24 | 79.82±5.54 | 0.44±0.10 |
| | DLinear | 70.46±4.65 | 70.34±4.63 | 70.56±4.64 | 64.84±4.31 | 0.41±0.09 |
| | TimesNet | 73.19±5.53 | 73.10±5.57 | 80.58±5.99 | 79.95±5.93 | 0.46±0.11 |
| | PathcTST | 76.80±4.73 | 76.73±4.77 | 84.94±4.98 | 83.87±5.42 | 0.54±0.09 |
| **OpenBMI** | TimesMixer | 54.08±4.38 | 48.91±8.99 | 54.62±6.35 | 54.03±5.04 | 0.08±0.09 |
| (2 Classes) | iTransfromer | 69.83±3.86 | 69.75±3.88 | 78.04±4.67 | 78.44±4.50 | 0.40±0.08 |
| | UniTS | 73.59±4.63 | 73.52±4.67 | 81.55±5.52 | 81.52±5.56 | 0.47±0.09 |
| | TimeMachine | 72.33±4.36 | 72.23±4.37 | 80.32±5.29 | 79.59±6.17 | 0.45±0.09 |
| | S-Mamba | 72.05±4.20 | 71.97±4.22 | 80.09±4.97 | 79.78±5.26 | 0.44±0.08 |
| | EEG and ECoG Models | | | | | |
| | Shallow ConvNet | 75.21±5.35 | 75.15±5.38 | 82.20±5.86 | 80.93±6.82 | 0.50±0.11 |
| | Deep ConvNet | 76.93±3.51 | 76.81±3.59 | 85.31±3.59 | 85.16±3.83 | 0.54±0.07 |
| | EEGNet | 74.93±4.48 | 74.80±4.52 | 83.20±5.11 | 82.71±5.26 | 0.50±0.09 |
| | TSception | 65.78±5.77 | 65.60±5.87 | 71.69±7.84 | 71.16±8.20 | 0.32±0.12 |
| | EEG Conformer | 77.90±5.27 | 77.78±5.33 | 85.81±5.23 | 85.36±5.47 | 0.56±0.11 |
| | Medformer | 74.06±3.35 | 73.99±3.37 | 81.65±4.05 | 81.38±4.38 | 0.48±0.07 |
| | **Corical-SSM (Ours)** | **81.62±5.20** (+3.72) | **81.56±5.24** (+3.78) | **90.14±4.61** (+4.33) | **90.17±4.47** (+4.81) | **0.63±0.10** (+0.07) |
| | Chance Performance | 25.00 | 25.00 | 50.00 | 25.00 | 0.00 |
| | General Time-Series Models | | | | | |
| | Informer | 59.45±7.70 | 58.32±8.69 | 83.30±6.03 | 67.17±10.36 | 0.46±0.10 |
| | Autoformer | 28.05±3.56 | 26.14±4.29 | 52.65±3.66 | 29.19±3.79 | 0.04±0.05 |
| | FEDformer | 27.34±3.06 | 26.17±3.63 | 52.87±3.08 | 29.46±3.27 | 0.03±0.04 |
| | Crossformer | 46.88±7.16 | 45.86±7.03 | 73.24±6.21 | 51.24±9.12 | 0.29±0.10 |
| | DLinear | 45.39±3.89 | 44.17±4.54 | 63.78±2.81 | 35.37±2.75 | 0.27±0.05 |
| | TimesNet | 57.50±9.57 | 55.88±10.48 | 81.37±6.51 | 63.32±11.32 | 0.43±0.13 |
| | PathcTST | 58.91±5.88 | 57.10±7.23 | 83.37±5.70 | 66.74±8.90 | 0.45±0.08 |
| **ECoG-ALS** | TimesMixer | 44.61±6.72 | 43.50±7.33 | 69.54±5.86 | 45.78±7.91 | 0.26±0.09 |
| (4 Classes) | iTransfromer | 56.09±8.74 | 53.84±11.43 | 81.59±5.38 | 64.09±10.34 | 0.41±0.12 |
| | UniTS | 48.52±13.57 | 44.72±17.84 | 74.89±11.59 | 53.85±15.72 | 0.31±0.18 |
| | TimeMachine | 56.09±6.10 | 55.40±6.40 | 78.69±5.06 | 58.88±7.50 | 0.41±0.08 |
| | S-Mamba | 53.28±7.35 | 52.31±8.15 | 78.71±5.45 | 59.16±9.19 | 0.38±0.10 |
| | EEG and ECoG Models | | | | | |
| | Shallow ConvNet | 44.45±7.16 | 41.53±17.97 | 77.10±7.51 | 51.98±8.52 | 0.26±0.10 |
| | Deep ConvNet | 59.30±12.15 | 56.94±14.91 | 85.73±8.52 | 70.68±13.48 | 0.46±0.16 |
| | EEGNet | 65.47±6.64 | 63.33±10.07 | 88.31±6.18 | 75.36±10.74 | 0.54±0.09 |
| | TSception | 46.56±13.76 | 44.85±14.61 | 73.10±12.27 | 51.71±16.69 | 0.29±0.18 |
| | EEG Conformer | 64.06±9.70 | 62.41±12.14 | 88.47±6.44 | 75.22±12.55 | 0.52±0.13 |
| | Medformer | 59.06±7.17 | 58.18±8.61 | 83.16±6.56 | 66.18±11.43 | 0.45±0.10 |
| | **Cortical-SSM (Ours)** | **75.08±9.69** (+9.61) | **73.87±12.84** (+10.54) | **92.34±6.40** (+3.87) | **83.25±12.34** (+7.89) | **0.67±0.13** (+0.13) |

90.14%, 90.17%, and 0.63 points, respectively. The proposed method outperformed the second-best model by 3.72, 3.78, 4.33, 4.81, and 0.07 points for these respective metrics. Moreover, the proposed method achieved state-of-the-art results on ECoG-ALS with accuracy, macro-F1, AUROC, AUPRC, and Cohen's Kappa of 75.08%, 73.87%, 92.34%, 83.25%, and 0.67 points, respectively. Compared with the second-best model, our method exhibited improvements of 9.61, 10.54, 3.87, 7.89, and 0.13 points for accuracy, macro-F1, AUROC, AUPRC, and Cohen's Kappa, respectively.

Furthermore, to compare performance across both benchmarks, we applied the Wilcoxon signed-rank test, a nonparametric test designed for paired samples. The results of this revealed statistically significant differences ($p < 0.05$) between our proposed method and the baseline methods across each evaluation metric and for both benchmarks. Additionally, we justified use of the above non-parametric test over the standard $t$-test through normality analysis. Specifically, we conducted Shapiro–Wilk tests on the accuracy values from each fold of the proposed method for both benchmarks, and these indicated rejection of the normality assumption ($p < 0.05$). This finding can be attributed to the inherent domain shifts across experimental sessions and subjects in EEG and ECoG signals (Cho et al., 2017; Kaya et al., 2018). These session- and subject-specific accuracy distributions violate the normality requirements of the parametric $t$-test, thereby necessitating nonparametric alternatives.

## 5.2 INTERPRETABILITY

We further examine our proposed method by generating visual explanations, following the procedure described in Appendix D. Figure 3 demonstrates sample-agnostic visual explanations of the proposed method on OpenBMI (Lee et al., 2019). Row (a) depicts temporal-frequency visualizations, whereas Rows (b) illustrate spatio-temporal visualizations. Columns (i) and (ii) show the class-wise average explanations for successful cases, corresponding to left-hand and right-hand grasp imagery, respectively. Specifically, Row (a) visualize $\mathbf{Z}_{\mathrm{freq}}^n$ (introduced in Appendix D), with 0 seconds indicating the onset of MI, while Row (b) visualize $\mathbf{Z}_{\mathrm{ch}}^n$ (also introduced in Appendix D) projected as topographic maps aligned with the 10–20 system (Klem, 1999). As seen in Row (a), the proposed method consistently attends to the mu band (approximately 10 Hz), a frequency range well known to be associated with MI in EEG (Pfurtscheller et al., 2006). Furthermore, Row (b) indicates that regions near electrodes C3 and C4 were strongly attended to. The C3 and C4 electrodes are positioned over the motor cortex and are neurophysiologically linked to hand motor control (Pfurtscheller et al., 2006). Therefore, these results suggest that the proposed method reliably attends to neurophysiologically critical regions in EEG, in a sample-agnostic manner.

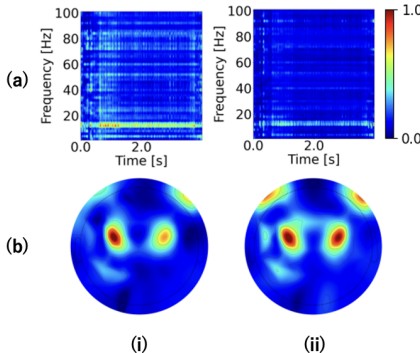

Figure 3: Sample-agnostic visual explanations of our proposed method on OpenBMI (Lee et al., 2019). Rows (a) and (b) display the temporal-frequency and spatio-temporal visual explanations for Subject 02, respectively. Columns (i) and (ii) correspond to left and right hand grasp, respectively.

Next, Figure 4 presents sample-agnostic visual explanations of our proposed method on ECoG-ALS. Columns (i)–(iv) display the class-wise average explanations corresponding to elbow extension, elbow flexion, hand extension, and hand grasp, respectively. Row (a) shows the average $\mathbf{Z}_{\mathrm{freq}}^n$ for successful cases in Session 07, aligned such that 0 seconds denotes the onset of MI. Row (b) visualizes the average $\mathbf{Z}_{\mathrm{ch}}^n$ for the same sessions, projected onto the subject's electrode configuration (as shown in Figure 5). As seen in Row (a), the proposed method consistently attends to a temporal interval of approximately 0.5 seconds centered around the MI onset. Moreover, Row (b) reveals that regions surrounding 9–25 and 49–60, located in the Hand Knob Area (Yousry et al., 1997) were consistently attended to. Previous studies have demonstrated that MI related to hand movements induces distinct increases in high-frequency components in ECoG signals from these regions. Accordingly, these results suggest that our proposed method effectively attends to neurophysiologically critical regions in a sample-agnostic manner, not only for EEG but also for ECoG.

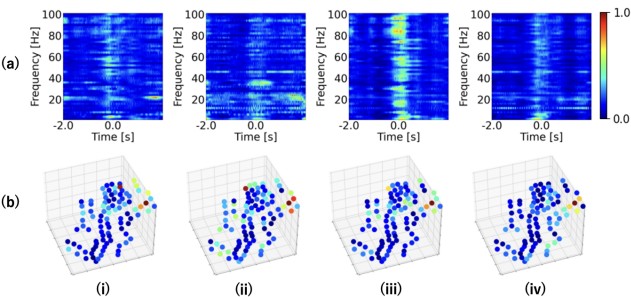

Figure 4: Sample-agnostic visual explanations of our proposed method on ECoG-ALS. Rows (a) and (b) display the temporal-frequency and spatio-temporal visual explanations for Session 07, respectively. Columns (i)–(iv) correspond to the output classes: (i) elbow extension, (ii) elbow flexion, (iii) hand extension, and (iv) hand grasp.

### 5.3 ABLATION STUDY

To investigate the effectiveness of each module, we conducted ablation studies on the following three conditions.

**Wavelet-Convolution ablation.** Table 2 presents the performance of different frequency-domain feature extraction methods in the E-Branch and A-Branch of the Wavelet-Convolution module. We compared five model variants (1-i)-(1-v) with combinations of short-time Fourier transform (STFT)

Table 2: Ablation study on frequency-domain feature extraction methods in the Wavelet-Convolution module. 1D-Conv. denotes 1D convolutional layers.

| Model | E-Branch | A-Branch | Accuracy [%] ↑ | |
| --- | --- | --- | --- | --- |
| | | | OpenBMI | ECoG-ALS |
| (1-i) | STFT | — | 71.51±3.94 | 45.31±6.30 |
| (1-ii) | CWT | — | 78.86±5.05 | 65.55±9.92 |
| (1-iii) | — | 1D-Conv. | 80.25±5.55 | 73.98±8.48 |
| (1-iv) | STFT | 1D-Conv. | 79.46±5.22 | 70.00±9.81 |
| (1-v) | CWT | 1D-Conv. | **81.62±5.20** | **75.08±9.69** |

and continuous wavelet transform (CWT) in the E-Branch, along with 1D convolutional layers in the A-Branch. The results reveal that models employing CWT in the E-Branch exhibited markedly different performance on OpenBMI (Lee et al., 2019), with the classification accuracy of Model (1-ii) being 2.76 points lower than that of Model (1-v). Similarly, when models using 1D convolutions in the A-Branch were compared, Model (1-iii) underperformed Model (1-v) by 1.37 points in the corresponding metric. These findings suggest that the complementary integration of deterministically obtained frequency components from the E-Branch and adaptively derived frequency features from the A-Branch enhances the model's representational capacity.

**Frequency-SSM and Channel-SSM ablation.** Table 3 shows the performance of different architectures in the Frequency-SSM and Channel-SSM. We compared models using the following architectures for capturing temporal dependencies: (2-i) Attention (Vaswani et al., 2017), (2-ii) S4-LegS (Gu et al., 2022), (2-iii) Mega (Ma et al., 2022), (2-iv) Mamba (Gu & Dao, 2024), (2-v) Mamba-2 (Dao & Gu, 2024), and (2-vi) S5 (Smith et al., 2023). Table 3 indicate that the classification accuracy of models (2-i), (2-ii), (2-iii), (2-iv), and (2-v) on OpenBMI underperformed Model (2-vi) by 2.88, 1.93, 1.76, 2,17, and 1.33 points, respectively. These results suggest that S5, a Deep SSM employing a time-invariant system and MIMO configuration, is effective for capturing temporal dependencies in EEG and ECoG signals.

Table 3: Ablation study on architectures for capturing temporal dependencies.

| Model | Architecture | Accuracy [%] ↑ | |
|---|---|---|---|
| | | **OpenBMI** | **ECoG-ALS** |
| (2-i) | Attention | 78.74±5.11 | 56.72± 9.37 |
| (2-ii) | S4-LegS | 79.69±4.98 | 69.38±11.22 |
| (2-iii) | Mega | 79.86±2.05 | 71.48±12.06 |
| (2-iv) | Mamba | 79.45±3.25 | 67.03± 8.56 |
| (2-v) | Mamba-2 | 80.29±3.69 | 68.36± 8.52 |
| (2-vi) | S5 | **81.62±5.20** | **75.08± 9.69** |

**Module-wise ablation.** Table 4 presents the performance of the three main modules: Wavelet-Convolution, Frequency-SSM, and Channel-SSM. We compared four model configurations: (3-i) exclusion of Wavelet-Convolution, (3-ii) exclusion of Frequency-SSM, (3-iii) exclusion of Channel-SSM, and (3-iv) the complete model incorporating all three modules. Regarding classification accuracy on OpnBMI, Models (3-i), (3-ii), and (3-iii) underperformed Model (iv) by 5.74, 2.48, and 0.98 points, respectively. These findings indicate that each module contributes to improving overall model performance, with the Wavelet-Convolution module exerting the most significant impact.

Table 4: Ablation study on the core modules.

| Model | Wavelet Conv. | Frequency SSM | Channel-SSM SSM | Accuracy [%] ↑ | |
|---|---|---|---|---|---|
| | | | | **OpenBMI** | **ECoG-ALS** |
| (3-i) | | ✓ | ✓ | 75.88±2.12 | 63.13±10.42 |
| (3-ii) | ✓ | | ✓ | 79.14±2.75 | 65.00± 8.93 |
| (3-iii) | ✓ | ✓ | | 80.64±2.31 | 73.91± 9.42 |
| (3-iv) | ✓ | ✓ | ✓ | **81.62±5.20** | **75.08± 9.69** |

## 6 CONCLUSION AND LIMITATIONS

**Conclusion.** In this study, we focused on a classification task based on EEG and ECoG signals recorded during MI tasks. We proposed Cortical-SSM, an extension of Deep SSMs designed to capture integrated dependencies across temporal, spatial, and frequency domains. For frequency feature extraction, we proposed the Wavelet-Convolution, which extracts non-black-box frequency-analyzable features while maintaining a learnable representation. In comprehensive evaluations on three MI EEG and ECoG benchmarks, our method consistently outperformed the comparison baseline methods. Furthermore, we demonstrated that for both EEG and ECoG signals, neurophysiologically significant regions were attended to in the visual explanations generated by our proposed method.

**Limitations and future work.** While Cortical-SSM captures integrated dependencies across temporal, spatial, and frequency domains from EEG and ECoG signals, it has limitations that suggest promising directions for future work. 1) Limited cross-domain integration: Cortical-SSM processes each domain independently to enhance interpretability, but this may cause overreliance on a single domain (e.g., temporal) when it suffices for the classification, underutilizing complementary cues. Future work could explore progressive or joint learning strategies to promote balanced feature utilization. See Appendix F.3 for detailed analysis. 2) Sensitivity to domain shifts: As with the existing baselines, Cortical-SSM does not explicitly address subject- or session-level domain shifts (Cho et al., 2017; Kaya et al., 2018). As shown in Section 5.1, the Shapiro–Wilk test rejects normality across folds, suggesting substantial variance. Incorporating domain adaptation techniques, such as learning invariant representations via mutual information maximization, offers a promising path forward.

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

## A  ADDITIONAL RELATED WORK

**Deep state space models.** Rooted in the theoretical foundations of classical state space models from control theory (Kalman, 1960), Deep SSMs leverage the Linear State-Space Layer (Gu et al., 2021) to model sequential data effectively. Moreover, a key innovation lies in the integration of the HiPPO matrix (Gu et al., 2020), which fuses the concept of recurrent memory with optimal polynomial approximation. Subsequent developments have substantially expanded this paradigm (Gu et al., 2022; Smith et al., 2023; Gu & Dao, 2024). For instance, S4 (Gu et al., 2022) introduced a learnable HiPPO parameterization, while S5 (Smith et al., 2023) extended the framework to a Multi-Input Multi-Output (MIMO) setting. Notably, the recent Mamba architecture (Gu & Dao, 2024) introduces an input-dependent selection mechanism that enables time-varying state transitions, and its performance is reported to surpass that of Transformers in certain language modeling tasks. This development has catalyzed cross-domain adoption, with success being demonstrated in fields such as computer vision (Zhu et al., 2025; Wang et al., 2025a), speech synthesis (Jiang et al., 2025; Li et al., 2024b), and medical applications (Ma et al., 2024; Guo et al., 2024). Together, these advancements establish Deep SSMs as a viable alternative to attention-based models.

**Multivariate time-series forecasting models.** Multivariate time-series forecasting has found broad applications across diverse domains, ranging from medical signal prediction (e.g., electromyography and electrocardiography analysis (Zhang et al., 2023; Li et al., 2024a)) as well as other applications (Kaneda et al., 2022; Bi et al., 2023; Huang et al., 2024). Notably, Transformer-based approaches are widely studied due to their demonstrated capability for long range sequence modeling (e.g., (Zhou et al., 2021; Wu et al., 2023; Nie et al., 2023; Zhang & Yan, 2023)). For example, Informer (Zhou et al., 2021) proposed an efficient forecasting method based on a ProbSparse self-attention mechanism and a generative decoder. Other notable examples include PatchTST (Nie et al., 2023), which divides time-series signals into smaller patches for Transformer input, and Crossformer (Zhang & Yan, 2023), which introduces a two-stage attention mechanism to capture inter-variable and temporal dependencies separately. In contrast, DLinear (Zeng et al., 2023) achieves performance comparable to Transformer-based approaches by utilizing a simple yet effective MLP-based method that decomposes time-series into seasonal and trend components. Drawing on this findings, iTransformer (Liu et al., 2024) proposes a strategy in which the matrix operations in the Transformer are transposed, thereby capturing inter-variable dependencies through the Attention mechanism while modeling temporal dependencies with an MLP.

**Datasets and benchmarks.** MI EEG/ECoG classification has been extensively studied using standardized benchmarks including BCI Competition III (Blankertz et al., 2006), BCI Competition IV (Brunner et al., 2008), and the dataset by Schalk et al. (Schalk et al., 2004). In MI EEG/ECoG, however, domain shifts across experimental sessions and subjects have been reported (Cho et al., 2017; Kaya et al., 2018; Maswanganyi et al., 2022). These domain shifts arise from intrinsic variability in neural signals, influenced by factors typified by participant's alertness and wakefulness during EEG/ECoG measurement. For instance, Maswanganyi et al. (Maswanganyi et al., 2022) observed approximately 12% accuracy variation in a four-class MI EEG classification task under cross-session evaluation. Similarly, Song et al. (Song et al., 2022) reported up to 41% performance discrepancies in cross-subject scenarios for another four-class MI-EEG classification task. While aforementioned benchmarks provide foundational insights, their limited number of experimental sessions and subjects hinders a comprehensive assessment of the impact of such domain shifts Therefore, in EEG settings, several multi-subject, multi-session benchmarks have been proposed including OpenBMI (Lee et al., 2019), as well as the datasets by Cho et al. (Cho et al., 2017), Kaya et al. (Kaya et al., 2018), and Stieger et al. (Stieger et al., 2021). In contrast, for ECoG settings, the substantial measurement barriers limit the feasibility of recording signals from multiple subjects, making the handling of domain shifts across experimental sessions a critical concern. To the best of our knowledge, the only standard benchmark comprising multiple sessions of ECoG signals is Task 1 of BCI Competition III, but it provides only a limited number of experimental sessions ($n = 2$). Therefore, in this work, we employ a clinical dataset of ECoG signals recorded from a single subject across multiple sessions ($n = 8$) during MI tasks.

## B  TERMINOLOGY

The terminology used in this paper is defined as follows:

- Electroencephalogram (EEG): Biosignals recorded from electrodes placed on the scalp.

- Electrocorticogram (ECoG): Biosignals recorded from electrodes placed on the brain surface following a craniotomy.

- Electrode: In EEG, this refers to wet electrodes or dry electrodes depending on the dataset, whereas in ECoG, this refers to subdural electrodes.

- Motor imagery (MI): Mental representation of physical movements without an overt motor output.

## C DEEP STATE SPACE MODELS

Recent advancements in Deep SSMs (Gu et al., 2022; Smith et al., 2023; Dao & Gu, 2024) have demonstrated their remarkable advantages over predominant architectures (including Transformer (Vaswani et al., 2017)) across various sequence modeling tasks. Inspired by classical SSM (Kalman, 1960), Deep SSMs establish a principled framework in which input signals $\mathbf{x}(t) \in \mathbb{R}^P$ are mapped to output signals $\mathbf{y}(t) \in \mathbb{R}^P$ via latent states $\mathbf{h}(t) \in \mathbb{R}^Q$, as follows:

$$\frac{d\mathbf{h}(t)}{dt} = \mathbf{A}\mathbf{h}(t) + \mathbf{B}\mathbf{x}(t), \qquad \mathbf{y}(t) = \mathbf{C}\mathbf{h}(t) + \mathbf{D}\mathbf{x}(t), \tag{8}$$

where $P$ and $Q$ denote the number of variables for the input/output signals and the latent states, respectively. Moreover, $\mathbf{A} \in \mathbb{R}^{Q \times Q}$ represents the state matrix, while $\mathbf{B} \in \mathbb{R}^{Q \times P}$, $\mathbf{C} \in \mathbb{R}^{P \times Q}$, and $\mathbf{D} \in \mathbb{R}^{P \times P}$ denote the projection matrices. Notably, the Deep SSM variant S5 (Smith et al., 2023) has demonstrated remarkable effectiveness in capturing sequential relationships for continuous signals. Here, we detail the processing steps employed in S5.

In S5, the HiPPO-N matrix (Gu et al., 2022) is adopted as $\mathbf{A}$ to effectively capture long-range dependencies in sequential signals. Since the HiPPO-N matrix is real symmetric, it can be diagonalized as $\mathbf{A} = \mathbf{V}\mathbf{\Lambda}\mathbf{V}^{-1}$, thereby transforming Equation (8) into the following form:

$$\frac{d\tilde{\mathbf{h}}(t)}{dt} = \mathbf{\Lambda}\tilde{\mathbf{h}}(t) + \tilde{\mathbf{B}}\mathbf{x}(t), \qquad \mathbf{y}(t) = \tilde{\mathbf{C}}\tilde{\mathbf{h}}(t) + \mathbf{D}\mathbf{x}(t), \tag{9}$$

where $\tilde{\mathbf{h}}(t) = \mathbf{V}^{-1}\mathbf{h}(t)$, $\tilde{\mathbf{B}} = \mathbf{V}^{-1}\mathbf{B}$, and $\tilde{\mathbf{C}} = \mathbf{C}\mathbf{V}$. Moreover, by introducing a timescale parameter $\mathbf{\Delta} \in \mathbb{R}_+$, Equation 9 is discretized using the zero-order hold (ZOH) method (Zhang & Chong, 2007) as follows:

$$\tilde{\mathbf{h}}_t = \bar{\mathbf{\Lambda}}\tilde{\mathbf{h}}_{t-1} + \bar{\mathbf{B}}\mathbf{x}_t, \qquad \mathbf{y}_t = \bar{\mathbf{C}}\tilde{\mathbf{h}}_t + \bar{\mathbf{D}}\mathbf{x}_t, \tag{10}$$

where $\bar{\mathbf{\Lambda}} = \exp(\mathbf{\Lambda}\mathbf{\Delta})$, $\bar{\mathbf{B}} = \mathbf{\Lambda}^{-1}\left(\bar{\mathbf{\Lambda}} - \mathbf{I}\right)\tilde{\mathbf{B}}$, $\bar{\mathbf{C}} = \tilde{\mathbf{C}}$, $\bar{\mathbf{D}} = \mathbf{D}$. In practice, $\mathbf{\Delta} \in \mathbb{R}^Q$ is used for the timescale parameter, and $\mathbf{D}$ is restricted as a diagonal matrix. Under these conditions, the learnable parameters consist of $\mathrm{diag}(\mathbf{\Lambda})$, $\tilde{\mathbf{B}}$, $\tilde{\mathbf{C}}$, $\mathrm{diag}(\mathbf{D})$, and $\mathbf{\Delta}$. Furthermore, S5 achieves efficient modeling of Equation 10 through the introduction of parallel scanning.

## D VISUAL EXPLANATIONS

In our proposed method, we generate visual explanations in both the spatio-temporal and temporal-frequency domains through the following procedures. Our approach extends Grad-CAM(Selvaraju et al., 2017) to generate visual explanations tailored for time-series signals. Grad-CAM is formulated as follows, generating a feature map $\mathbf{Z}^n \in \mathbb{R}^{I \times J}$ that represents the explanation for the $n$-th output class from the gradient of the loss with respect to $\hat{y}_n$:

$$\alpha_r = \frac{1}{IJ} \sum_{i \in I} \sum_{j \in J} \frac{\partial \hat{y}^{(n)}}{\partial o_{r,i,j}}, \qquad \mathbf{Z}^{(n)} = \mathrm{ReLU}\left(\sum_{r \in R} \alpha_r \mathbf{o}_r\right), \tag{11}$$

where $\hat{y}_n$ denotes the $n$-th element of $\hat{\mathbf{y}}$. $I$, $J$, and $R$ denote the number of vertical and horizontal pixels, and the number of dimensions of the feature map, respectively. Furthermore, $\mathbf{o}_r \in \mathbb{R}^{I \times J}$ and

$o_{r,i,j} \in \mathbb{R}$ denote the feature map of the $r$-th dimension and the feature indexed by position $(i,j)$ within $\mathbf{o}_r$, respectively. In this context, $\mathbf{o}_r$ is required to retain the spatial relationships of the input.

Next, we describe our method for generating visual explanations in the spatio-temporal domain. While Grad-CAM operates on 2D feature maps $\mathbf{o}_r$ and performs weighting across $\{\mathbf{o}_r\}_{r=1}^R$, our method deals with time-series signals and therefore performs weighting across 1D time-series vectors. Here, we generate visual explanations using $\mathbf{V}^{(L)}$, the feature closest to the output layer, which preserves the spatio-temporal relationships of the input signal. As detailed in Subsection 3.4, the feature within $\mathbf{V}^{(L)}$ are processed independently for each electrode. Therefore, we generate a separate visual explanation for each electrode by weighting across $\left\{\mathbf{v}_{f,m}^{(L)}\right\}_{f=1}^F$. This yields the feature map $\mathbf{Z}_{\mathrm{ch}}^n \in \mathbb{R}^{M \times T}$, representing the spatio-temporal visual explanation for the $n$-th output class, obtained as follows:

$$\alpha_{f,m}^n = \frac{1}{T} \sum_{t \in T} \frac{\partial \hat{y}_n}{\partial v_{f,m,t}^{(L)}}, \tag{12}$$

$$\mathbf{Z}_{\mathrm{ch}}^n = \left\{ \mathrm{ReLU}\left( \sum_{f \in F} \alpha_{f,m}^n \mathbf{v}_{f,m}^{(L)} \right) \,\middle|\, m = 1 \ldots M \right\}. \tag{13}$$

Finally, we describe our method for generating visual explanations in the temporal-frequency domain. Here, we generate visual explanations using $\mathbf{U}^{(L)}$, the feature closest to the output layer, which preserves the temporal-frequency relationships of the input signal. As detailed in Subsection 3.3, in contrast to $\mathbf{V}^{(L)}$, the features within $\mathbf{U}^{(L)}$ are processed independently for each frequency component. Therefore, we generate a separate visual explanation for each frequency component by weighting across $\left\{\mathbf{u}_{f,m}^{(L)}\right\}_{m=1}^M$. This yields the feature map $\mathbf{Z}_{\mathrm{freq}}^n \in \mathbb{R}^{F \times T}$, representing the temporal-frequency visual explanation for the $n$-th output class, as in Equations 12 and 13.

# E    DETAILED EXPERIMENTAL SETUP

## E.1    DATASETS

We constructed the training, validation, and test sets according to the procedures described below, using them for model training, hyperparameter tuning, and performance evaluation, respectively. During training, we computed classification accuracy on the validation set after each epoch. For the final evaluation on the test set, we employed the model parameters corresponding to the highest validation accuracy.

**OpenBMI.** It is a public dataset of EEG recorded from 54 healthy subjects while they performed an MI task (Lee et al., 2019). The task involves two types of MI: right-hand grasping and left-hand grasping. The experimental design was based on the protocol established by Pfurtscheller et al. (Pfurtscheller & Neuper, 2001). While recording the signals, a display was positioned in front of the subjects, and the type movement to be imagined was indicated through visual cues including left and right arrows.

Subjects engaged in two sessions, performing 400 MI trials per session. A total of 21,600 samples were collected from 54 subjects. Each sample was recorded at a sampling rate of 1000 Hz, following a 3-second presentation of fixation cross, a 4-second MI, and a 6-second ($\pm 1.5$ seconds) rest period. Additionally, visual cues were continuously presented during the 4-second MI period. EEG signals were recorded using BrainAmp systems with Ag–AgCl electrodes, and a total of 62 electrodes were positioned according to the international 10–20 system (Klem, 1999). Notably, this study focused on the 4-second MI period.

We adopted $k$-fold cross-validation ($k = 8$) in a subject-independent manner. EEG samples collected from 54 subjects were partitioned at the subject level and randomly shuffled across the subjects. For each fold, samples from 44, 5, and 5 subjects were allocated to the training, validation and test sets, respectively. Given that each subject contributed 400 samples, the training, validation, and test sets consist of 17600, 2000, and 2000 samples, respectively.

**Stieger2021.** It is a public dataset of EEG recorded from 62 healthy subjects during motor imagery (Stieger et al., 2021). This dataset consists of three types of tasks(LR, UD, and 2D; defined below) using MI control a cursor on a display. The movements to be imagined included left-hand grasping, right-hand grasping, bilateral hand grasping, and voluntarily rest, with each task involving different combinations of these movements. In the LR task, left-hand and right-hand grasping correspond to cursor movements in the left and right directions, respectively. The UD task involved bilateral hand grasping and voluntarily rest, which correspond to cursor movements in the upward and downward directions, respectively. The 2D task integrated the LR and UD tasks, which involved four types of movements, with cursor movement directions corresponding to those in the LR and UD tasks. While recording the signals, a display was positioned in front of the subjects, and a yellow bar indicating the direction corresponding to the movement was presented at the edge of the screen as a visual cue. Subjects were required to move the cursor, located at the center of the screen, by imagining the movement corresponding to the direction of the yellow bar. In this process, cursor movement was controlled based on the alpha power from electrodes C4 and C3 in the international 10–10 system (Nuwer, 2018).

Subjects engaged in 7 to 11 sessions, performing MI tasks ranging from 1050 to 1650 trials per task. A total of 269099 samples were collected from 64 subjects. Each sample was recorded at a sampling rate of 1000 Hz, following a 2-second rest period, a 2-second presentation of visual cues, and up to 6 seconds of MI. The MI recording was terminated when the cursor reached the screen edge in any direction. EEG signals were recorded using SynAmps RT and Neuroscan caps, with 64 electrodes positioned according to the international 10–10 system. Notably, this study focused on 202950 samples obtained from 41 participants who completed all 11 sessions. For each sample, we used a 4 second window, encompassing 2 seconds before and 2 seconds after the onset of MI.

Similar to OpenBMI, we employed $k$-fold cross validation ($k = 8$) in a subject-independent manner. EEG samples from 41 subjects were first partitioned at the individual level and randomly shuffled among the subjects. Subsequently, samples from 33, 4, and 4 subjects were allocated to the training, validation, and test sets, respectively. With each subject providing 1650 samples, the dataset was split into training, validation, and test sets of 54450, 6600, and 6600 samples, respectively.

**ECoG-ALS.** It is an ECoG dataset recorded from a single subject with ALS during an MI task. This task involved four types of MI: elbow extension and flexion, as well as hand extension and grasping. While recording the signals, a display was positioned in front of the subject, and the type of movement to be imagined was communicated via textual instructions. To ensure precise timing of MI, synchronized auditory and visual cues, including voice prompts and on-screen arrows, were presented to the subject.

A single subject performed the four types of MI, each repeated 40 times per experimental session, over the course of eight sessions. Consequently, a total of 1280 samples were collected from the subject. Each sample was recorded at a sampling rate of 1000 Hz following a 4-second rest and 2-seconds of MI. In this study, we used a 4-second window encompassing 2 seconds before and 2 seconds after the onset of MI. The ECoG signals

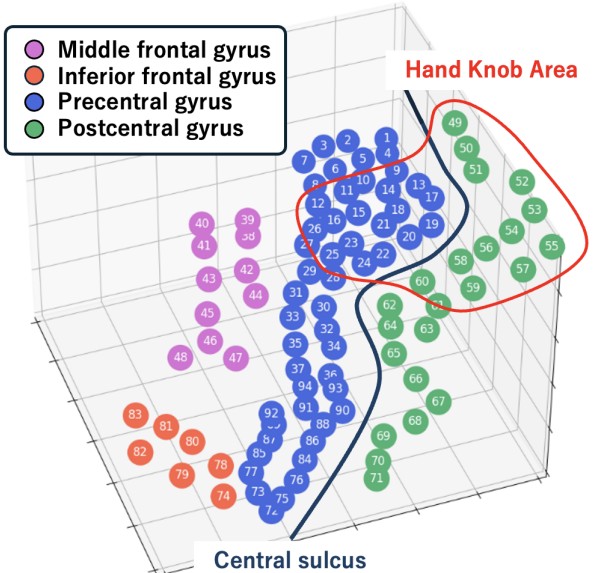

Figure 5: Electrode distribution in ECoG-ALS. Electrodes 9–25 and 49–60, positioned within the precentral and postcentral gyrus, respectively, are located in the Hand Knob Area (Yousry et al., 1997).

were recorded from 94 subdural electrodes positioned near the central sulcus of the left hemisphere. Figure 5 illustrates the spatial distribution of these electrodes. Among the electrodes positioned in the precentral and postcentral gyrus, electrodes 9–25 and 49–60 were located within the Hand

Table 5: Performance comparison for the LR and UD tasks in Stieger2021 (Stieger et al., 2021). **Bold** and underlined text indicate the best and second-best methods for each evaluation metric.

| Dataset | Models | Accuracy [%] ↑ | Macro-F1 [%] ↑ | AUROC [%] ↑ | AUPRC [%] ↑ | Kappa ↑ |
|---|---|---|---|---|---|---|
| | Chance Performance | 50.00 | 50.00 | 50.00 | 50.00 | 0.00 |
| | *General Time-Series Models* | | | | | |
| | Informer | 77.90±5.51 | 77.85±5.54 | 86.83±5.87 | 87.05±5.91 | 0.56±0.11 |
| | Autoformer | 72.02±5.02 | 71.96±5.02 | 79.75±6.08 | 79.22±6.25 | 0.44±0.10 |
| | FEDformer | 67.48±4.05 | 67.16±4.51 | 74.46±5.02 | 73.36±5.34 | 0.35±0.08 |
| | Crossformer | 82.63±4.81 | 82.60±4.83 | 91.80±4.03 | 92.09±3.89 | 0.65±0.10 |
| | DLinear | 63.66±4.37 | 63.65±4.36 | 63.70±4.39 | 58.94±3.42 | 0.27±0.09 |
| | TimesNet | 77.43±5.98 | 77.41±5.98 | 86.37±6.36 | 86.54±6.46 | 0.55±0.12 |
| Stieger2021 | PathcTST | 79.67±3.52 | 79.64±3.52 | 88.85±3.76 | 88.47±4.75 | 0.59±0.07 |
| **LR Task** | TimesMixer | 53.97±4.88 | 46.91±9.21 | 54.29±5.09 | 53.33±3.86 | 0.08±0.10 |
| (2 Classes) | iTransfromer | 78.67±5.77 | 78.62±5.82 | 87.85±5.93 | 88.08±6.00 | 0.57±0.12 |
| | UniTS | 82.59±3.05 | 82.58±3.05 | 91.21±2.90 | 91.38±2.90 | 0.65±0.06 |
| | TimeMachine | 79.23±5.83 | 79.22±5.83 | 87.17±5.93 | 86.30±6.44 | 0.58±0.12 |
| | S-Mamba | 80.23±5.70 | 80.21±5.71 | 88.67±5.73 | 88.67±5.73 | 0.60±0.11 |
| | *EEG and ECoG Models* | | | | | |
| | Shallow ConvNet | 52.22±2.68 | 42.95±7.18 | 64.44±3.90 | 60.60±3.58 | 0.04±0.05 |
| | Deep ConvNet | 69.56±2.97 | 69.15±2.86 | 75.01±6.09 | 70.68±6.67 | 0.39±0.06 |
| | EEGNet | 75.16±5.92 | 75.06±5.93 | 79.75±6.95 | 75.57±7.31 | 0.50±0.12 |
| | TSception | 63.19±6.38 | 60.29±9.98 | 70.46±3.00 | 67.02±2.99 | 0.26±0.13 |
| | EEG Conformer | 72.53±5.73 | 72.42±5.77 | 79.90±6.92 | 77.89±7.53 | 0.45±0.11 |
| | Medformer | 78.67±6.12 | 78.61±6.15 | 88.06±5.87 | 88.47±5.75 | 0.57±0.12 |
| | **Cortical-SSM (Ours)** | **87.12±4.33** (+4.49) | **87.11±4.33** (+4.51) | **94.66±3.40** (+1.86) | **94.82±3.32** (+2.73) | **0.74±0.09** (+0.09) |
| | Chance Performance | 50.00 | 50.00 | 50.00 | 50.00 | 0.00 |
| | *General Time-Series Models* | | | | | |
| | Informer | 72.55±5.23 | 72.46±5.26 | 81.18±6.26 | 81.33±6.44 | 0.45±0.10 |
| | Autoformer | 63.49±2.95 | 62.81±3.41 | 69.28±4.11 | 67.87±4.24 | 0.27±0.06 |
| | FEDformer | 64.23±2.77 | 63.83±2.94 | 69.73±3.19 | 68.03±3.13 | 0.28±0.06 |
| | Crossformer | 76.96±4.71 | 76.74±5.01 | 86.19±4.28 | 86.27±4.34 | 0.54±0.09 |
| | DLinear | 54.92±2.49 | 54.72±2.67 | 54.96±2.48 | 52.80±1.47 | 0.10±0.05 |
| | TimesNet | 72.14±4.63 | 72.09±4.62 | 80.33±5.16 | 80.26±5.31 | 0.44±0.09 |
| Stieger2021 | PathcTST | 73.90±3.74 | 73.86±3.74 | 82.47±4.45 | 81.99±5.21 | 0.48±0.07 |
| **UD Task** | TimesMixer | 52.32±2.72 | 44.48±8.01 | 53.19±3.00 | 52.55±2.31 | 0.05±0.05 |
| (2 Classes) | iTransfromer | 64.94±6.48 | 64.84±6.54 | 71.11±8.39 | 70.54±8.89 | 0.30±0.13 |
| | TimeMachine | 72.92±5.17 | 72.88±5.20 | 80.55±5.87 | 79.78±6.20 | 0.46±0.10 |
| | S-Mamba | 74.58±4.77 | 74.50±4.82 | 82.67±5.37 | 82.47±5.47 | 0.49±0.10 |
| | *EEG and ECoG Models* | | | | | |
| | Shallow ConvNet | 52.13±3.70 | 43.00±8.90 | 61.16±3.78 | 58.38±3.43 | 0.04±0.07 |
| | Deep ConvNet | 54.38±3.93 | 46.26±9.81 | 64.26±4.21 | 61.41±3.43 | 0.09±0.08 |
| | EEGNet | 51.34±2.44 | 38.23±4.78 | 66.48±6.17 | 63.17±5.57 | 0.03±0.05 |
| | TSception | 52.95±4.09 | 44.15±10.11 | 57.94±6.08 | 56.43±4.94 | 0.06±0.08 |
| | EEG Conformer | 66.97±6.98 | 66.02±8.06 | 73.91±8.01 | 73.01±8.37 | 0.34±0.14 |
| | Medformer | 74.14±4.97 | 74.02±5.01 | 83.08±5.07 | 83.37±5.07 | 0.48±0.10 |
| | **Cortical-SSM (Ours)** | **79.09±4.16** (+2.13) | **79.03±4.17** (+2.29) | **87.97±4.08** (+1.78) | **88.06±4.16** (+1.79) | **0.58±0.08** (+0.04) |

Knob Area (Yousry et al., 1997). Previous studies have reported that this area exhibits a significant increase in high-frequency components during motor imagery (MI), supporting the relevance of these electrodes to the current task.

For the ECoG-ALS dataset, we employed $k$-fold cross-validation ($k = 8$) in a session-independent manner. ECoG samples collected over eight experimental sessions on different days were segmented per session and randomly shuffled across the sessions. Subsequently, samples from six sessions were assigned to the training set, one session was assigned to the validation set, and the remaining session was assigned to the test set. Given that there are 160 samples per session, the training, validation, and test sets consisted of 960, 160, and 160 samples, respectively.

## E.2 IMPLEMENTATION DETAILS

We employed the AdamW optimizer ($\beta_1 = 0.9, \beta_2 = 0.999$) with a learning rate of $1.0 \times 10^{-4}$. The batch size was set to 8 and the training ran for 100 epochs. For the Wavelet-Convolution, we set the frequency dimension $F = 50$ and targeted the frequency range $(f_{\min}, f_{\max}) = (1 \text{ Hz}, 100 \text{ Hz})$. For both the Frequency-SSM and Channel-SSM, we set the stacked block number $L = 2$.

Overall, our model contained approximately 0.93 million trainable parameters and 2.34 billion multiply-add operations. Training was performed on a Nvidia GeForce RTX 4090 with 24GB of

Table 6: Performance comparison for the 2D task in Stieger2021 (Stieger et al., 2021). **Bold** and underlined text indicate the best and second-best methods for each evaluation metric.

| Dataset | Models | Accuracy [%] ↑ | Macro-F1 [%] ↑ | AUROC [%] ↑ | AUPRC [%] ↑ | Kappa ↑ |
|---|---|---|---|---|---|---|
| | Chance Performance | 25.00 | 25.00 | 50.00 | 25.00 | 0.00 |
| | *General Time-Series Models* | | | | | |
| | Informer | 55.92±7.68 | 55.69±7.90 | 80.93±5.97 | 64.10±9.41 | 0.41±0.10 |
| | Autoformer | 39.55±3.66 | 39.28±3.81 | 65.81±3.63 | 40.25±4.53 | 0.19±0.05 |
| | FEDformer | 38.02±2.77 | 37.85±2.89 | 64.14±2.81 | 37.82±3.14 | 0.17±0.04 |
| | Crossformer | 62.88±7.03 | 62.92±7.16 | 86.02±4.69 | 72.60±8.09 | 0.51±0.09 |
| | DLinear | 33.96±3.42 | 33.85±3.47 | 56.06±2.30 | 28.36±1.60 | 0.12±0.05 |
| | TimesNet | 54.01±7.78 | 53.62±8.14 | 79.48±6.08 | 61.40±9.48 | 0.39±0.10 |
| Stieger2021 | PathcTST | 59.27±7.33 | 58.98±7.51 | 82.84±5.22 | 66.52±8.64 | 0.46±0.10 |
| 2D Task | TimesMixer | 26.72±2.57 | 18.14±7.63 | 51.66±2.67 | 26.24±1.85 | 0.02±0.03 |
| (4 Classes) | iTransfromer | 56.37±7.85 | 56.23±7.84 | 81.12±6.18 | 64.23±9.79 | 0.42±0.10 |
| | UniTS | 61.90±3.11 | 61.84±3.10 | 85.16±2.10 | 70.50±3.72 | 0.49±0.04 |
| | TimeMachine | 58.21±7.30 | 58.11±7.33 | 81.76±5.68 | 64.72±8.79 | 0.44±0.10 |
| | S-Mamba | 60.27±7.70 | 60.20±7.69 | 83.20±5.82 | 67.38±9.29 | 0.47±0.10 |
| | *EEG and ECoG Models* | | | | | |
| | Shallow ConvNet | 27.43±2.09 | 17.77±4.48 | 59.26±1.65 | 31.72±1.83 | 0.03±0.03 |
| | Deep ConvNet | 26.45±2.32 | 15.29±3.80 | 65.09±3.24 | 36.54±3.72 | 0.02±0.03 |
| | EEGNet | 28.85±4.11 | 20.38±5.34 | 68.55±5.31 | 41.16±6.17 | 0.05±0.05 |
| | TSception | 25.90±1.35 | 15.96±3.66 | 53.26±1.91 | 27.05±1.51 | 0.01±0.02 |
| | EEG Conformer | 55.76±7.40 | 55.27±7.47 | 79.77±5.92 | 61.25±9.83 | 0.41±0.10 |
| | Medformer | 57.69±7.95 | 57.64±7.90 | 82.20±5.67 | 66.14±9.06 | 0.44±0.11 |
| | **Cortical-SSM (Ours)** | **66.07±7.34** (+3.19) | **66.01±7.40** (+3.09) | **87.67±4.66** (+1.65) | **75.37±8.17** (+2.77) | **0.55±0.10** (+0.04) |

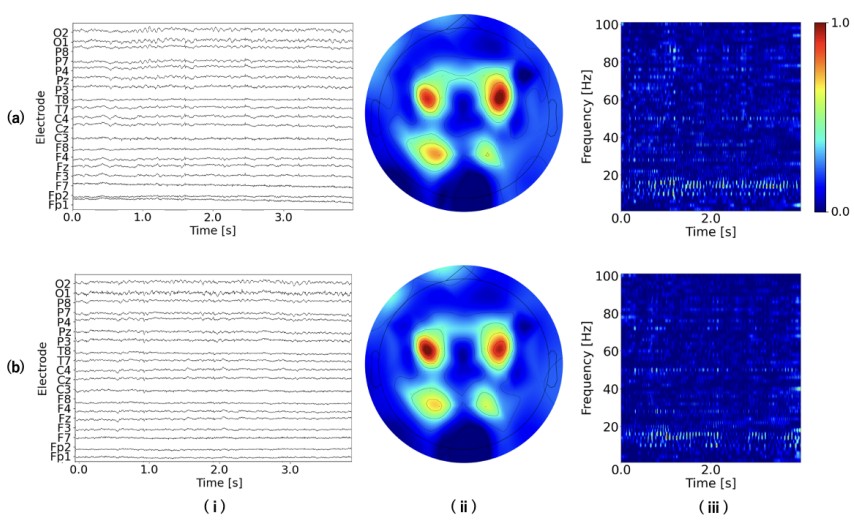

Figure 6: Successful cases of the proposed method on OpenBMI (Lee et al., 2019). Rows (a) and (b) display samples for which the ground-truth class is left-hand grasping. For each sample, Column (i) shows recorded EEG signals, Column (ii) presents the spatio-temporal visual explanations, and Column (iii) illustrates the temporal-frequency visual explanations.

VRAM and an Intel Core i9-13900KF processor with 64GB of RAM. The training times per task for our proposed method on OpenBMI, Stieger2021, and ECoG-ALS were approximately 157 min, 50 min, and 4 min, respectively. The inference time per sample in the corresponding benchmarks was approximately 2-3 ms.

# F  ADDITIONAL RESULTS

## F.1  QUANTITATIVE RESULTS FOR STIEGER2021

Tables 5 and 6 present the quantitative results of the proposed method and the baseline methods for the three tasks (LR, UD, and 2D) included in Stieger2021(Stieger et al., 2021). For details on the selection of the baseline methods and evaluation metrics, please refer to Section 5.1. According to

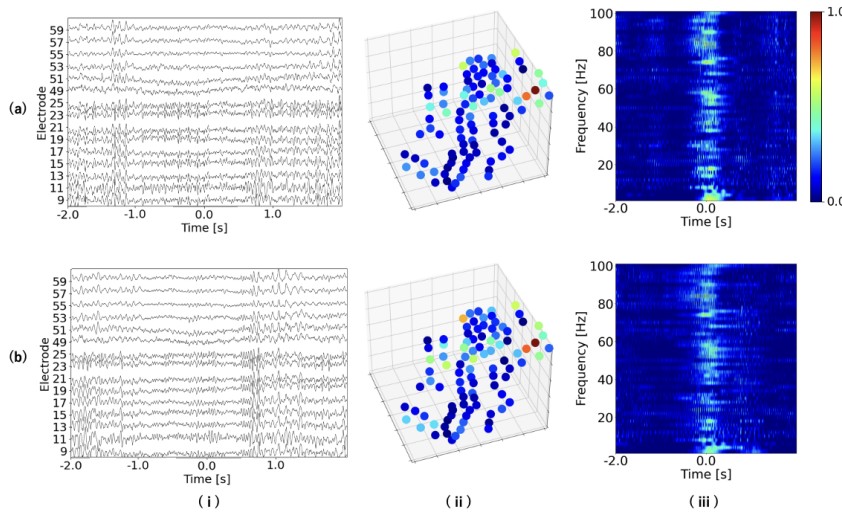

Figure 7: Successful cases of the proposed method on ECoG-ALS. In this figure, Rows (a) and (b) correspond to samples whose ground-truth class is hand extension. For each sample, Column (i) displays the recorded ECoG signals, Column (ii) presents the spatio-temporal visual explanations, and Column (iii) illustrates the temporal-frequency visual explanations.

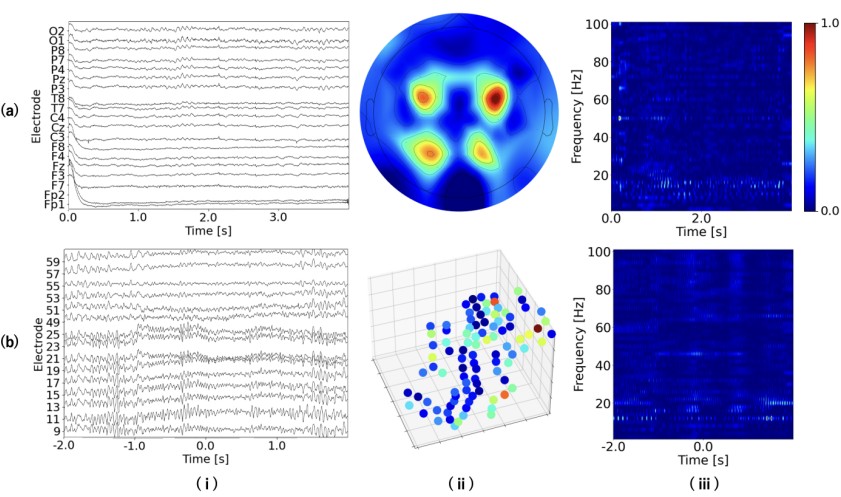

Figure 8: Failure cases of the proposed method. Rows (a) and (b) show samples from OpenBMI (Lee et al., 2019) and ECoG-ALS, respectively. Columns (i)–(iii) present, respectively, (i) the recorded EEG or ECoG signals, (ii) the visual explanations in the temporal-frequency domain, and (iii) the visual explanations in the spatio-temporal domain.

Table 6, the proposed method achieved the highest performance in the 2D task of Stieger2021, with accuracy, macro-F1, AUROC, AUPRC, and Cohen's Kappa of 66.07%, 66.01%, 87.67%, 75.37%, and 0.55 points, respectively. Similarly, as shown in Tables 5 and 6, the proposed method also demonstrated the best performance in both the LR and UD tasks of Stieger2021.

## F.2 SAMPLE-SPECIFIC INTERPRETABILITY

Figure 6 shows successful cases by the proposed method on OpenBMI (Lee et al., 2019). In the figure, Rows (a) and (b) display samples where the ground truth class corresponds to left-hand grasping, while Columns (i)-(iii) respectively present: (i) recorded EEG signals, (ii) spatio-temporal visual explanations, and (iii) temporal-frequency visual explanations. Column (ii) reveals that the proposed method consistently attended to regions surrounding the C3 and C4 electrodes. Furthermore, in contrast to the sample-agnostic visual explanations in Figure 3, the proposed method in this sample also attended to regions near the P3 and P4 electrodes, located over the left and right parietal cortices.

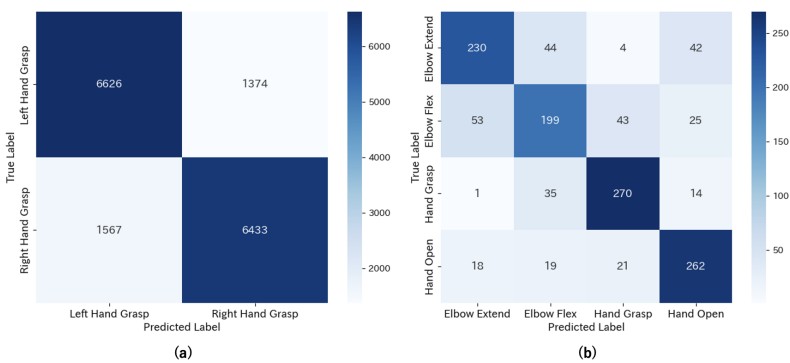

Figure 9: Confusion matrices for OpenBMI (Lee et al., 2019) and ECoG-ALS.

These areas are associated with somatosensory processing (Pause et al., 1989; Dijkerman & de Haan, 2007) and may hold functional relevance. Since the OpenBMI dataset includes motor imagery of both left- and right-hand movements, it is plausible that these regions capture proprioceptive representations of the imagined limb movements. Additionally, Column (iii) indicates consistent attention to regions surrounding the mu band. Taken together, these results suggest that the proposed method effectively focuses on neurophysiologically meaningful EEG regions.

Next, Figure 7 presents successful cases of the proposed method on ECoG-ALS. In the figure, Rows (a) and (b) display samples where the ground truth class is hand extension. Following the structure of Figure 6, Columns (i)-(iii) respectively present: (i) recorded ECoG signals, (ii) spatio-temporal visual explanations, and (iii) temporal-frequency visual explanations. As shown in Row (ii), electrodes located in the Hand Knob Area (Yousry et al., 1997) were significantly attended. Meanwhile, Row (iii) indicates that the period approximately 0.5 seconds around the MI onset was strongly attended to. Therefore, these results suggest that our proposed method effectively attends to neurophysiologically critical regions of ECoG.

Finally, Figure 8 illustrates cases where the proposed method failed. Row (a) shows a sample from OpenBMI where the ground-truth label was left-hand grasp, yet our method incorrectly predicted right-hand grasp. Similarly, Row (b) presents a sample from ECoG-ALS in which the ground-truth label was hand extension, but the method erroneously predicted elbow extension. Columns (i)–(iii) correspond, respectively, to the (i) recorded EEG or ECoG signals, (ii) the temporal-frequency visual explanations, and (iii) the spatio-temporal visual explanations. Notably, from Subfigures 8 (a-ii) and 8 (b-ii), regions surrounding the C3/C4 electrodes and the Hand Knob Area were significantly attended to. In contrast, Subfigures 8 (a-iii) and 8 (b-iii) indicate that the spatio-temporal domain was weakly attended to. These observations suggest that, for these samples, the proposed method predominantly focused on neurophysiologically critical regions in the spatial domain. We hypothesize that this misclassification can be attributed to domain shifts across experimental sessions and subjects in EEG, and across experimental sessions in ECoG, with the impact of domain shift in the spatial domain being relatively minor compared with that in the temporal-frequency domain.

## F.3 ERROR ANALYSIS

Figure 9 presents the confusion matrices for all test sets obtained through $k$-fold cross validation ($k = 8$) for (a) OpenBMI (Lee et al., 2019) and (b) ECoG-ALS. For OpenBMI, the F1-scores for left and right hand grasp were 81.84% and 81.39%, respectively. In contrast, for ECoG-ALS, the elbow extension class achieved 230 True Positives (TP), 72 False Positives (FP), 888 True Negatives (TN), and 90 False Negatives (FN), resulting in an F1-score of 73.95%. Likewise, the F1-scores for the elbow flexion, hand extension, and hand grasp classes were 64.51%, 82.07%, and 79.03%, respectively. Therefore, the hand extension class exhibited the highest F1-score. On average, there were 367.63 failure modes per fold for OpenBMI and 39.88 per fold for ECoG-ALS.

To investigate the limitations of our proposed method, we conducted an error analysis on 100 failure modes from both OpenBMI and ECoG-ALS. The causes of failure can be broadly grouped into three categories:

Table 7: Error analysis on failure modes

| Error | OpenBMI | ECoG-ALS |
|---|---|---|
| Irrelevant Attention Error (IAE) | 51 | 72 |
| Prediction Error (PE) | 23 | 23 |
| Noise/Artifact Induced Error (NAIE) | 26 | 5 |
| Total | 100 | 100 |

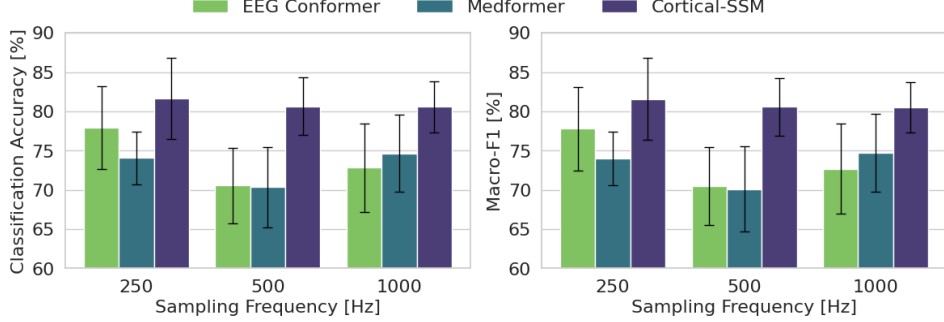

Figure 10: Sensitivity analysis of EEG sequence length on OpenBMI (Lee et al., 2019).

- **Irrelevant Attention Error (IAE)**: This category refers to modes where regions that deviate from neurophysiologically significant areas were attended to, or where regions in the temporal, spatial, and frequency domains were insufficiently attended.

- **Prediction error (PE)**: This category includes cases modes where neurophysiologically significant regions were attended to, yet the predictions were incorrect.

- **Noise/artifact Induced Error (NAIE)**: This category refers to modes where the EEG or ECoG signal-to-noise ratio (SNR) was below -15 dB. For SNR calculation, we defined 1-100 Hz as the signal band, and 0-1 Hz and above 100 Hz as noise bands.

Furthermore, we defined neurophysiologically significant regions as follows:

- OpenBMI: The mu band around 10 Hz and regions surrounding the C3 and C4 electrodes in the 10–20 system (Klem, 1999).

- ECoG-ALS: Regions near 0 seconds representing the onset of MI and areas surrounding the Hand Knob Area (Yousry et al., 1997).

Table 7 indicates that the main bottleneck in both benchmarks was IAE. We hypothesize that this issue arises because our method tends to overemphasize certain regions within the temporal, spatial, and frequency domains during training. Our approach processes features from each domain independently to provide explicit visual explanations from all three domains. Consequently, when successful classification is achieved using features from only a subset of these domains, features from the remaining domains may not be effectively utilized. A possible solution could be training with features from the temporal, spatial, and frequency domains in a stepwise manner, which would be effective for achieving balanced attention across domains and reducing IAE. Furthermore, the number of NAIE modes was 26 for OpenBMI and 5 for ECoG-ALS, revealing a substantial difference between the benchmarks. This disparity can be attributed to the distinct recording methodologies of EEG and ECoG signals. The lower NAIE in ECoG signals likely results from their proximity to the signal source, leading to higher signal quality compared with EEG signals.

## F.4 SENSITIVITY ANALYSIS ON SEQUENCE LENGTH

To investigate the effect of sequence length on decoding performance, we performed a sensitivity analysis on the OpenBMI dataset by resampling EEG signals originally recorded at 1000 Hz to three sampling rates (250 Hz, 500 Hz, and 1000 Hz), as illustrated in Figure 10. The 250 Hz condition

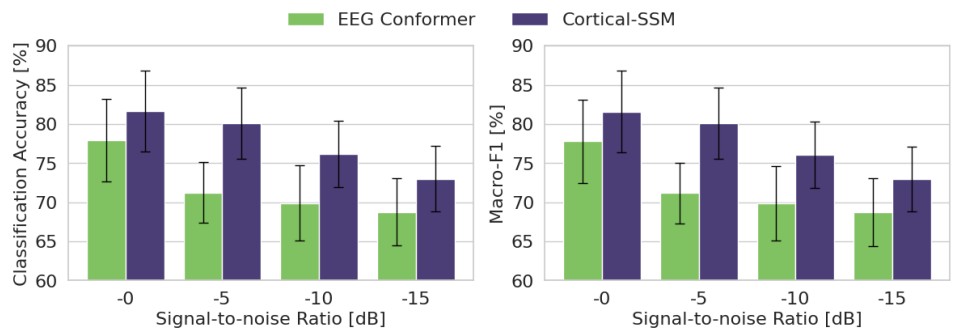

Figure 11: Sensitivity analysis of EEG signal-to-noise ratio on OpenBMI (Lee et al., 2019).

corresponds to the setting used in the original experimental setup, and all conditions employ 4-second EEG segments as input. We compared our method against two representative EEG/ECoG decoding baselines: EEG Conformer (Song et al., 2022) and Medformer (Wang et al., 2024d). While EEG Conformer and Medformer revealed accuracy drops of up to 7.34 and 3.73 points, respectively, our method shows only a marginal decrease of 1.06 points. These results suggest that the proposed method scales with sequence length.

### F.5 SNESIIVITY ANALYSIS ON SIGNAL-TO-NOISE RATIO

To evaluate the impact of noise and signal degradation, we conducted an additional experiment in which the signal-to-noise ratio (SNR) of EEG signals from the OpenBMI dataset was progressively degraded by 0dB, 5dB, 10dB, and 15dB as shown in Figure 11. Here, we compared the performance of our proposed method with the second-best baseline, EEG Conformer (Song et al., 2022). Following Appendix F.3, the signal bandwidth was set to 1-100Hz. As shown in the figure, our method consistently outperformed EEG Conformer across all SNR levels in terms of classification accuracy and macro-F1, demonstrating its effectiveness even in the presence of substantial noise and signal degradation common in clinical settings.

## G THE USE OF LARGE LANGUAGE MODELS

In this work, large language models (LLMs) were used solely for auxiliary purposes, such as polishing, editing, and formatting the manuscript. They were not employed in any part of the methodology, nor did they contribute to the implementation or evaluation of the proposed methods. Therefore, LLMs were not involved as essential, original, or non-standard components of the research.

