# OpenReview forum: "Cortical-SSM: A Deep State Space Model for EEG and ECoG Motor Imagery Decoding"
_ICLR.cc/2026/Conference — Submitted to ICLR 2026_

### Official Review · Reviewer_DVQH · 2025-10-25

**Soundness:** 3
**Presentation:** 3
**Contribution:** 2
**Rating:** 4
**Confidence:** 3

**Summary:**

The authors propose Cortical-SSM, extending deep state space models to capture integrated dependencies of EEG and ECoG signals across temporal, spatial, and frequency domains. The manuscript is well-written and presents extensive content. However, the concern is the relatively limited technical contribution of the work.

**Strengths:**

[1] Apply deep state space models to EEG MI\
[2] Two tasks: EEG MI and ECoG MI

**Weaknesses:**

[1] Wavelet-Convolution – The wavelet transformation with convolution/CNN has been applied in the current literature, such as:\
Hou et al., A novel approach of decoding EEG four-class motor imagery tasks via scout ESI and CNN, In Journal of Neural Engineering.\
The authors are encouraged to compare the performance under different convolution settings, e.g., pure convolution, convolution in the frequency domain, etc.\
[2] Equation (1) – For convolution, batch normalization is usually used, instead of layer normalization. Why did the authors apply a layer normalization after the wavelet transform?\
[3] Experimental datasets – The authors are encouraged to employ the largest benchmark in the EEG MI field: \
EEG Motor Movement/Imagery Dataset\
Link: https://archive.physionet.org/pn4/eegmmidb/ \
High Gamma Dataset\
Link: https://github.com/robintibor/high-gamma-dataset

**Questions:**

[1] How was the visual explanation in Figure 3 related to the human neuron activations with the EEG signals?\
[2] How was the performance of Wavelet-Convolution, compared with the convolution (the second term in Equation (1))? And why didn’t the authors conduct convolution after the Wavelet transform?\
[3] What are the specific technical contributions on the SSM, e.g., any architecture innovations or new algorithm design? Or, did the authors just simply apply the SSM to EEG MI?\
[4] Where and what is the interpretation of the interpretability/visual maps?

---

### Official Review · Reviewer_5YxK · 2025-11-01

**Soundness:** 2
**Presentation:** 2
**Contribution:** 2
**Rating:** 2
**Confidence:** 4

**Summary:**

The paper proposes Cortical-SSM, a state-space model for EEG and ECoG motor-imagery decoding. It combines a wavelet-based spectral extraction module with dual SSMs (frequency- and channel-wise) to model temporal, spatial, and spectral dependencies, and reports improved performance on three datasets.

**Strengths:**

The integration of a wavelet–convolution frequency module with state-space models applied along the frequency and channel domains offers a unified treatment of temporal, spatial, and spectral dependencies in EEG/ECoG, representing a transfer of recent SSM advances to neural signal modeling. Empirically, the paper reports consistent improvements across both EEG and ECoG benchmarks.
This cross-domain application is interesting and demonstrates that SSMs can be effective for modeling long-term temporal dependencies in neural signals.

**Weaknesses:**

The approach largely follows the conventional EEG pipeline (temporal–spatial–spectral features) and appears to recombine existing components; the core modules (CWT, SSM) are adopted without modification or a clear theoretical contribution.

The interpretability claim rests on a straightforward use of Grad-CAM, which by itself may not constitute a methodological advance.

The Wavelet-Convolution combines handcrafted and learnable filters with a simple weighted-sum fusion, but this conventional design is not well justified nor clearly shown to improve spectral representation or interpretability.

For ECoG, the evidence is limited to a single-subject ALS dataset, which constrains the strength and generalizability of the conclusions.

The comparative evaluation would benefit from broader baselines, including recent general time-series models and state-of-the-art EEG/ECoG approaches [1-3].

[1] Rusch, T. Konstantin, and Daniela Rus. "Oscillatory State-Space Models." In Proceedings of the Thirteenth International Conference on Learning Representations (ICLR), 2025.
[2] Wang, Jiquan, Sha Zhao, Zhiling Luo, Yangxuan Zhou, Haiteng Jiang, Shijian Li, Tao Li, and Gang Pan. “CBraMod: A Criss-Cross Brain Foundation Model for EEG Decoding.” In Proceedings of the Thirteenth International Conference on Learning Representations (ICLR), 2025.
[3] Chen, Xupeng, et al. "A neural speech decoding framework leveraging deep learning and speech synthesis." Nature Machine Intelligence 6.4 (2024): 467-480.

The manuscript asserts lower computational cost and improved preservation of fine-grained temporal dependencies. However, the manuscript includes no relevant comparisons or analyses to substantiate these claims.

**Questions:**

Do Figures 3–4 present only averaged success cases, and if so, how do the authors address possible selection bias? Have they examined failure cases or conducted quantitative evaluation?

Could the authors clarify why frequency extraction relies solely on the Morlet wavelet, and whether they considered comparisons with other wavelets (e.g., Gabor [1], Morse [2]) or with learnable filters (e.g., SincNet [3])? Such comparisons would help determine whether the reported gains stem from the overall design or from the specific frequency filter used [1-3].
[1] Luan, Shangzhen, et al. "Gabor convolutional networks." IEEE Transactions on Image Processing 27.9 (2018): 4357-4366.
[2] Olhede, Sofia C., and Andrew T. Walden. "Generalized morse wavelets." IEEE Transactions on Signal Processing 50.11 (2002): 2661-2670.
[3] Ravanelli, Mirco, and Yoshua Bengio. "Speaker recognition from raw waveform with sincnet." 2018 IEEE spoken language technology workshop (SLT). IEEE, 2018.

The introduction jumps directly from the limitations of Transformer-based approaches to your proposed SSM, without explaining why SSM is a natural or necessary alternative. Could you elaborate on the specific properties of SSMs (e.g., linear recurrence, stability, computational scaling) that make them particularly suitable for EEG/ECoG signals?

Stieger2021 dataset appears only in the Appendix, even though it could provide an important validation point. Is there a reason this dataset was not emphasized in the main text?

Figure 2 is hard to interpret. Could you clarify the flow of information across modules and specify what input each SSM receives?

---

### Official Review · Reviewer_WUtg · 2025-11-01

**Soundness:** 3
**Presentation:** 3
**Contribution:** 3
**Rating:** 6
**Confidence:** 2

**Summary:**

This paper proposes Cortical-SSM model, which is a deep state-space architecture for motor-imagery (MI) decoding from EEG and ECoG. The model has three components: Wavelet-Convolution that fuses the deterministic continuous wavelet transform (CWT) features (E-branch) with learnable Conv1D features (A-branch), Frequency-SSM that models spatio-temporal dynamics per frequency band, and Channel-SSM that models temporal-frequency dynamics per electrode. Both SSM stacks build on S5 (time-invariant, MIMO). The method is evaluated on OpenBMI, Stieger2021, and a single-subject ECoG-ALS dataset with cross-subject or cross-session protocols. The authors report SOTA results on all three datasets with statistically significant gains via Wilcoxon tests, and provide Grad-CAM–style visualizations highlighting mu-band, C3/C4 in EEG, and the hand-knob area in ECoG.

**Strengths:**

1.	The proposed Vortical model can capture dependencies of EEG and ECoG across temporal, spatial, and frequency modals.
2.	This paper is presented with solid cross-dataset evidence. On OpenBMI, and on ECoG-ALS it reaches obviously better performance than previous baselines. Stieger2021 (LR/UD/2D) also shows consistent improvements.
3.	The authors make ablation study, they test STFT vs CWT, and different temporal backbones, and use Wilcoxon with Shapiro–Wilk to justify non-parametric testing under session/subject shift.
4.	The method can also capture the neurophysiological relevant regions of EEG and ECoG signals, providing better explanation ability than other methods.

**Weaknesses:**

1. The novelty is incremental. The method primarily recombines known ingredients of S5-style SSMs for long sequences, wavelet features for EEG, and Grad-CAM for explanations, into a two-branch temporal pipeline. The core technical advance is largely architectural refactoring rather than a new modeling principle or learning objective.

2. The ECoG generalization is unclear. ECoG evidence is based on a single ALS subject across eight sessions. Cross-session gains are strong but between-subject generalization and electrode-layout variability central for clinical translation are not evaluated.

3. The task of classification of EEG and ECoG signals for motor imagery is an extensively researched one, why the authors did not consider to use the model for intermediate representation of EEG and ECoG, so that the learned representation can be used to more downstream tasks.

**Questions:**

Please refer to the weaknesses part.

---

### Official Review · Reviewer_8g7z · 2025-11-03

**Soundness:** 3
**Presentation:** 3
**Contribution:** 2
**Rating:** 4
**Confidence:** 5

**Summary:**

This paper proposed a method based on state-space models to capture EEG and ECoG features for motor imagery classification tasks. Representations were extracted from temporal, spatial, and frequency domains to achieve good classification performance and interpretability, while the Wavelet-convolution was emphasized for frequency features. Two datasets, containing non-invasive EEG and invasive ECoG, were used for comparison.

**Strengths:**

1. Introduce a new state-space model for EEG and ECoG feature extraction.
2. Clear performance on different datasets and model ablation studies was reported to demonstrate the superior ability of the proposed model.
3. The authors paid attention to brain feature analysis, which gave evidence of the usability of the Cortical-SSM model.
4. The writing is easy to follow.
5. The appendix gave many details which is useful to know the model and reproduction.

**Weaknesses:**

1. The summary of the contributions claimed at the end of the introduction has not been well-proven. Although the work achieved good performance on various datasets, it's still not easy to figure out which part of the model really contributes to the final results, and what the improvement is compared to related works, such as DeepSSM mentioned in the paper.
2. There have been some analyses to show the spatial and temporal patterns obtained by the classification models. However, up to the novelty mentioned in the introduction, the current visualization of one subject or one session (Fig. 3 and Fig. 4) is not enough to prove the claim.
3. There are uncertain points of the pattern analysis: 1) the motor imagery of left and right in Fig. 3 showed similar patterns, but there should be some synchronization and desynchronization. 2) The frequency band of Column 3 in Fig. 4(a) spanned across all the frequency bands. Would that be some artifacts? 3) It is not clear what the content is shown in Fig. 4(b). Maybe further visualization of different subjects or the average one would show rich patterns.
4. The modules of the model did not seem to have a significant improvement for EEG data in Tables 3 and 4. Why did they have a large impact on the ECoG data? Some explanation would be beneficial for understanding.

**Questions:**

1. Have you evaluated the performance under domain shift conditions, such as cross-subject, cross-session?
2. Would the parameters of the Cortical-SSM have an impact on the overall performance?
3. Please see the Weakness.

---

### Meta-Review · Area_Chair_twiL · 2026-01-06

**Summary:**

The paper proposes a state-space model for motor imagery classification. Frequency-wise and channel-wise SSMs and wavelet-based spectral extraction are two main components. Evaluation was done on two datasets.

Strength: (1) The model seems to be able to capture spectral, temporal, and spatial dependencies in EEG/ECoG.
(2) The proposed method shows good performance consistently on the two datasets.
(3) The presentation quality is good.

Weakness: (1) All reviewers raises concerns regarding novelty/contribution.
(2) ECoG evaluation is limited to only one dataset.
(3) Several concerns/questions are raised by the reviewers regarding the results, analysis, and model design, implying that the paper lacks clarity.
(4) The claim about lower computational cost is not supported by results.

**Reviewer Concerns:**

No rebuttal was submitted.

**Reviewer Scores:**

The cores wouldn't have been changed through reviewer discussion. The reviewers with high confidence are all on the negative side.

---

### Decision · Program_Chairs · 2026-01-26

Reject